# The effect of a six-month programme of intradialytic cycling on survival and hospitalisations in people requiring haemodialysis: 5-year follow-up of the CYCLE-HD randomised controlled trial

Sherna F. Adenwalla[1,2], Hannah M. Worboys[3,4,5], Daniel Lawday[4], Laura J. Gray[3,4,5,6], Katherine L. Hull [1,2], Darren R. Churchward[1,2], Patrick J. Highton[5,7], Hannah M. L. Young[3,7], Matthew P. M. Graham-Brown[1,2,4], James O. Burton [1,2,4], Daniel S. March [1,2,4]*

1 Department of Cardiovascular Sciences, University of Leicester, Leicester United Kingdom, 2 Department of Renal Medicine, University Hospitals of Leicester NHS Trust, Leicester United Kingdom, 3 Department of Population Health Sciences, University of Leicester, Leicester, United Kingdom, 4 NIHR Leicester Biomedical Research Centre, University of Leicester and University Hospitals of Leicester NHS Trust, Leicester, United Kingdom, 5 NIHR Applied Research Collaboration East Midlands, University of Leicester, Leicester, United Kingdom, 6 Leicester British Heart Foundation Centre of Research Excellence, University of Leicester, Leicester, United Kingdom, 7 Leicester Diabetes Centre, College of Life Sciences, University of Leicester, Leicester, United Kingdom

* dsm12@leicester.ac.uk

## Abstract

We have previously shown that a six-month programme of intradialytic cycling (IDC) improved cardiovascular structure and function, it is unclear whether these changes are associated with long-term benefits. The aim of this post-trial analysis was to evaluate a programme of IDC on all-cause mortality, hospitalisations and cardiovascular events at five-years. Mortality and hospitalisation data were collected from Hospital Episode Statistics and death certificates. Models were fitted unadjusted and adjusted for age, sex, diabetes, duration of dialysis, and receiving a kidney transplant. Cox proportional hazard models were used for time-to-event analysis to evaluate all-cause mortality. Hospitalisations were analysed using a negative binomial regression model, and length of stay using a generalised linear model. A composite outcome of time to first cardiovascular event, combining cardiovascular mortality and hospitalisations, was evaluated using a Cox model. There was no evidence of a statistically significant effect of treatment allocation on survival (hazard ratio (HR) 1.09, 95% confidence interval (CI): 0.68–1.76, $p = 0.71$). After adjustment, results remained non-significant (HR 1.22, 95% CI: 0.74–2.01, $p = 0.43$). There was no evidence of a significant effect on all-cause hospitalisations for unadjusted ($p = 0.20$) or adjusted (p = 0.25) models. Similar results are reported for cardiovascular hospitalisations ($p = 0.30$ and $p = 0.17$). For time to first cardiovascular event there was no evidence of a statistically

**Data availability statement:** Data is available at the following link: https://doi.org/10.25392/leicester.data.29721590.

**Funding:** The CYCLE-HD study was independent research funded by the National Institute for Health and Care Research (NIHR) in the United Kingdom (grant reference number CS-2013-13-014; JOB) and supported by Kidney Research UK. Professor James O Burton was recipient of the funding award. The funders had no role in study design, data collection and analysis, decision to publish, or preparation of the manuscript.

**Competing interests:** JOB and LJG are both funded (Senior Investigator Awards) by the National Institute for Health and Care Research (NIHR). LG and HW are funded by the National Institute for Health and Care Research (NIHR) Applied Research Collaboration East Midlands (ARC EM) and Leicester NIHR Biomedical Research Centre (BRC). The views expressed are those of the author(s) and not necessarily those of the NIHR or the Department of Health and Social Care. For the purpose of open access, the author has applied a Creative Commons Attribution license (CC BY) to any Author Accepted Manuscript version arising from this submission.

significant effect (HR 1.39, 95% CI: 0.79–2.72, $p = 0.26$). The main findings show no evidence that a six-month programme of IDC affected all-cause mortality, hospitalisations, cardiovascular events, or length of stay in hospital at five-years.

## Introduction

Individuals with end stage kidney disease (ESKD) requiring haemodialysis have a significantly increased risk of cardiovascular disease [1], which is one of the leading causes of death in this population [2]. Traditional strategies to mitigate cardiovascular risk are less effective than in the general population, as pathophysiological processes are driven by clustering of both traditional and non-traditional risk factors such as chronic inflammation, uraemia, anaemia, renal bone disease and haemodialysis-induced myocardial stunning [3–5]. These processes drive pathological remodelling, resulting in cardiomyopathy, impaired left ventricular function and myocardial fibrosis [3].

Short-term studies have shown that exercise modifies some of the pathological cardiovascular changes that associate with poor outcomes [6]. Consequently, it is recommended that individuals receiving haemodialysis are physically active and that this may reduce risk of cardiovascular-related and all-cause mortality [7]. This is supported by longitudinal cohort studies showing that mortality levels are lower in those who self-report being more physically active [8,9]. For these reasons, the guideline recommendations for physical activity are graded as strong, whilst the evidence is rated as low quality, as a result of the study designs of the evidence base [7]. There is a lack of appropriately powered randomised controlled trials (RCTs) assessing the effect of exercise on outcomes such as hospitalisation and mortality in the haemodialysis population. That said, two previous RCTs investigating a 12-month intradialytic exercise programme [10], and a six-month walking programme [11,12] in the dialysis population have reported a reduction in hospitalisations [10], and a reduction in a combined endpoint of hospitalisation and mortality at 36 months [11].

We have previously shown that a six-month programme of exercise improved cardiovascular structure and function [6], and previous research has shown that the benefits of exercise can persist for a number of years after its cessation [11,13]. Therefore, the aim of this post-trial analysis was to assess the effect of this six-month programme of intradialytic cycling (IDC) on five-year all-cause mortality, cardiovascular mortality, and all-cause and cardiovascular hospitalisations.

## Methods

Demographic data for this study were taken from baseline data for 130 participants recruited to the CYCLE-HD study (ISRCTN11299707); an RCT investigating the effect of six-months of IDC (on top of usual dialysis care) on cardiac structure and function, assessed by cardiac magnetic resonance imaging (MRI), compared to usual dialysis care only. Inclusion and exclusion criteria, and the sample size calculation underlying the primary study are as previously described [6,14]. The study received

ethical approval by the NHS Research Ethics Committee East Midlands (Northampton; REC ref: 14/EM/1190) which allowed for collection of this data. All participants provided written informed consent. The original study recruited from 15th March 2015–7th of March 2018. Following study completion, participants in the intervention arm ceased IDC.

For this post-hoc, longitudinal analysis, original CYCLE-HD participants were followed up for five years from baseline (or beginning of the time horizon). The beginning of time-horizon was baseline study assessment for the control group and beginning of intervention for the IDC group. These time horizons were chosen because this ensures that it included the six-month IDC programme for the IDC group, and allows direct comparison with the control group. Data regarding mortality and hospitalisations (admission and discharge date and primary diagnosis at discharge) were collected from the University Hospitals of Leicester Hospital Episode Statistics (HES) database using structured query language. Cause of death was obtained from death certificates and categorised according to the UK Kidney Association's Renal Registry report (cardiac disease, cerebrovascular disease, infection, malignancy, treatment withdrawal, other, uncertain aetiology) [15]. Hospitalisations associated with a primary cardiovascular diagnosis were defined using ICD-10 criteria, as specified in S1 Table.

## Outcomes

The primary outcome for the present study was all-cause mortality at five years. Secondary outcomes included cardiovascular mortality, all-cause hospitalisations, cardiovascular hospitalisations, and length of stay in hospital. Due to a small number of cardiovascular events, a composite outcome was created which included cardiovascular death and cardiovascular hospitalisations.

## Statistical analysis

All statistical analyses were conducted using Stata software version 18.0 (Stata Corporation, College Station, Texas, USA). Normally distributed data were expressed as mean ± standard deviation and non-normally distributed data were expressed as median (interquartile range). Hypothesis tests were two-sided, with p-values <0.05 considered significant. All models were fitted unadjusted and adjusted for pre-specified explanatory variables; age, sex, diabetes, duration of dialysis, and receiving a transplant since trial enrolment (these are all known to modify risk of mortality in this population). Receiving a transplant was treated as a time-varying covariate, done by reshaping the dataset into two rows per transplanted participant, each corresponding to a time interval where the transplant covariate remains constant. This allows participants to contribute different risk periods before and after receiving a transplant. Survival was described using the Kaplan-Meier method. The primary analysis used Cox regression model of all-cause survival by treatment allocation (either control or IDC). Participants were censored at five years post-randomisation, or withdrawal of consent. The proportional hazards assumption for all Cox-regression models was tested using Schoenfeld residuals [16]. The total number of hospitalisations was analysed as a count variable using a negative binomial regression model. This model can account for a right skew in the distribution of the outcome data. Length of stay in hospital was analysed using a generalised linear model (GLM). Total hours were analysed as a continuous variable using a gamma distribution and log link function to account for the right skew in the outcome data. Due to the limited number of cardiovascular deaths (n = 24) and total cardiovascular hospitalisations (n = 59), a composite outcome was constructed to assess the time to the first cardiovascular event. Cardiovascular event as a composite outcome was evaluated using a cox proportional hazard model for time to first event. Participants were censored at five years post-randomisation, upon withdrawal of consent, or upon death due to a non-cardiovascular cause.

## Results

The results from the primary study have been published elsewhere [6]. A total of 130 participants were enrolled and completed baseline assessments, with 65 participants in each group. Baseline demographics for the IDC group and control group are shown in Table 1. 130 patients completed baseline assessments. During the six-month intervention

**Table 1. Baseline demographics for the CYCLE-HD cohort.**

|  | Control group (*n*=65) | Intradialytic cycling (IDC) group (*n*=65) |
|---|---|---|
| Age (years) | 58.9±14.9 | 55.5±15.5 |
| Male sex (%) | 53 (82%) | 42 (65%) |
| Dialysis vintage (years) | 1.3 [0.4, 3.2] | 1.2 [0.5, 3.7] |
| Ethnicity |  |  |
| White | 28 (43%) | 30 (46%) |
| Mixed | 0 (0%) | 2(3%) |
| Asian or Asian British | 29 (45%) | 24 (37%) |
| Black or Black British | 5 (8%) | 5 (8%) |
| Other | 3 (5%) | 4 (6%) |
| Pre-dialysis SBP (mmHg) | 143.0±20.3 | 143.1±23.3 |
| Pre-dialysis DBP (mmHg) | 75.1±13.6 | 77.1±14.2 |
| Haemoglobin (g/L) | 112 [99, 122] | 113 [105, 122] |
| Albumin (g/L) | 37±5.2 | 36.8±4.7 |
| Total Cholesterol (mmol/L) | 3.8±1.1 (*n*=50) | 4.2±1.6 (*n*=52) |
| Triglycerides (mmol/L) | 1.56 [0.97, 2.7] (*n*=45) | 1.46 [0.96, 1.83] (*n*=38) |
| HbA1c (%) | 5.8 [5.1, 7.1] (*n*=54) | 5.4 [4.9, 6.7] (*n*=38) |
| Pre-dialysis weight (kg) | 77.3±17.2 | 80.6±20.8 |
| *Medications* |  |  |
| Total weekly dose EPO (Units) | 7192±6087 | 7179±7845 |
| Total weekly dose IV Iron (mg) | 228±782 | 212±659 |
| ACEi/ARB | 12 (19%) | 16 (25%) |
| Beta-Blocker | 35 (54%) | 37 (57%) |
| Calcium Channel Blockers | 29 (45%) | 29 (45%) |
| Diuretics | 17 (26%) | 10 (15%) |
| *Co-morbidities* | (*n*=65) | (*n*=64) |
| Ischemic heart disease | 9 (14%) | 7 (11%) |
| Hypertension | 44 (68%) | 42 (66%) |
| Diabetes mellitus | 28 (43%) | 21 (33%) |
| Atrial Fibrillation | 2 (3%) | 3 (5%) |
| Previous renal transplant | 11 (17%) | 9 (14%) |

Data presented as mean ± SD, or median [25th, 75th percentile] or *n* (%). ace inhibitor (ACEi), angiotensin II receptor blocker (ARB), diastolic blood pressure (DBP), erythropoietin (EPO), intravenous (IV), systolic blood pressure (SBP).

period, 7 (5%) patients withdrew their consent and were censored at the point of withdrawal. During the data collection time-horizons (including the six-month intervention period (for the IDC group) and follow up), 47 (36%) participants received a kidney transplant (control *n*=25; IDC group *n*=22), 71 (55%) participants died (control *n*=37; IDC group *n*=34). There were 24 (18%) cardiovascular deaths (control *n*=9; IDC group *n*=15), 1,038 all-cause hospitalisations (control *n*=566; IDC group *n*=472), and 59 cardiovascular hospitalisations (control *n*=35; IDC group *n*=24). This data is summarised in Table 2.

## All-cause mortality

Participants were followed for a total of 486.603 patient-years. In the primary analysis, there was no statistically significant effect of treatment allocation on survival (hazard ratio [HR] for all-cause mortality 1.09 95% confidence interval (CI)

Table 2. 5-year follow-up data for the CYCLE-HD cohort for mortality, hospitalisations and a composite outcome of cardiovascular death or CARDIOVASCULAR hospitalisations.

| | Control group (n=65) | Intradialytic cycling (IDC) group (n=65) | Total (n=130) |
|---|---|---|---|
| **Mortality** | | | |
| Total number of all-cause deaths (%) | 37 (57%) | 34 (52%) | 71 (55%) |
| Total number of cardiovascular deaths (%) | 9 (14%) | 14 (22%) | 23 (18%) |
| **Hospitalisations** | | | |
| Total number of all-cause hospitalisations (n) | 566 | 472 | 1038 |
| Total number of hours spent in hospital for all-cause hospitalisations (n) | 85,510 | 66,076 | 151,586 |
| Total number of cardiovascular hospitalisations (n) | 35 | 24 | 59 |
| Total number of hours spent in hospital for cardiovascular hospitalisations (hours) | 15,732 | 6,348 | 22,080 |
| Number of individuals who had at least one cardiovascular event (n) | 22 | 26 | 48 |
| Number of individuals who received a kidney transplant (n) | 23 | 22 | 45 |

0.68–1.76, $p=0.71$) (Fig 1, Table 3). After adjustment, the results remained non-significant (HR 1.22, 95% CI: 0.74–2.01, $p=0.43$). Age and diabetes were the only significant predictors of mortality, with a 4% increased risk in mortality for each additional year of age ($p<0.01$), and a 66% increased risk of mortality for those with diabetes compared to those without ($p=0.03$) (Table 3).

### All-cause hospitalisations

There was no statistically significant effect of treatment allocation on the number of all-cause hospitalisations for unadjusted ($p=0.20$) or adjusted ($p=0.25$) models. Similar results are reported for cardiovascular hospitalisations ($p=0.30$ and $p=0.17$) (Table 4). Overall, total time spent in hospital was 151,586 hours, 85,510 hours in the control arm and 66,076 hours in the IDC arm. There was no statistically significant effect of treatment allocation on the length of stay in hospital for the unadjusted ($p=0.64$) or adjusted ($p=0.50$) models. Similar results are reported for length of stay for cardiovascular hospitalisations ($p=0.34$ and $p=0.83$) (Table 4).

### Composite outcome

For the time to the first cardiovascular event (a composite of cardiovascular deaths and cardiovascular hospitalisations), there was no statistically significant effect of treatment allocation to IDC (HR 1.39, 95% CI: 0.79–2.72, $p=0.26$) (Table 3). The Kaplan-Meier curve for time to first cardiovascular event is shown in Fig 2. After adjustment for age, sex, diabetes status, and dialysis vintage, and transplant status, the results remained non-significant (HR 1.52, 95% CI: 0.84–2.46, $p=0.26$). Age, diabetes and transplant status were identified as significant predictors of cardiovascular events, with a 2% increased risk per additional year of age ($p=0.08$), an 89% increased risk associated with diabetes ($p=0.03$). For the time varying covariate transplant, a hazard ratio of 0.18 means that there is a lower risk of the event after transplant. In this analysis this means receiving a transplant is associated with a 72% decreased risk of death, compared to before having a transplant ($p=0.02$) (Table 3).

### Discussion

This study investigated five-year follow-up of an RCT testing the effect of a six-month IDC programme. The main findings were that a six-month programme of IDC did not affect all-cause mortality, cardiovascular events (a composite of cardiovascular deaths and hospitalisations), all-cause hospitalisations or hospital length of stay. Age and diabetes were identified as significant determinants of cardiovascular events, with a 3.2% increased risk per additional year of age, and a 89%

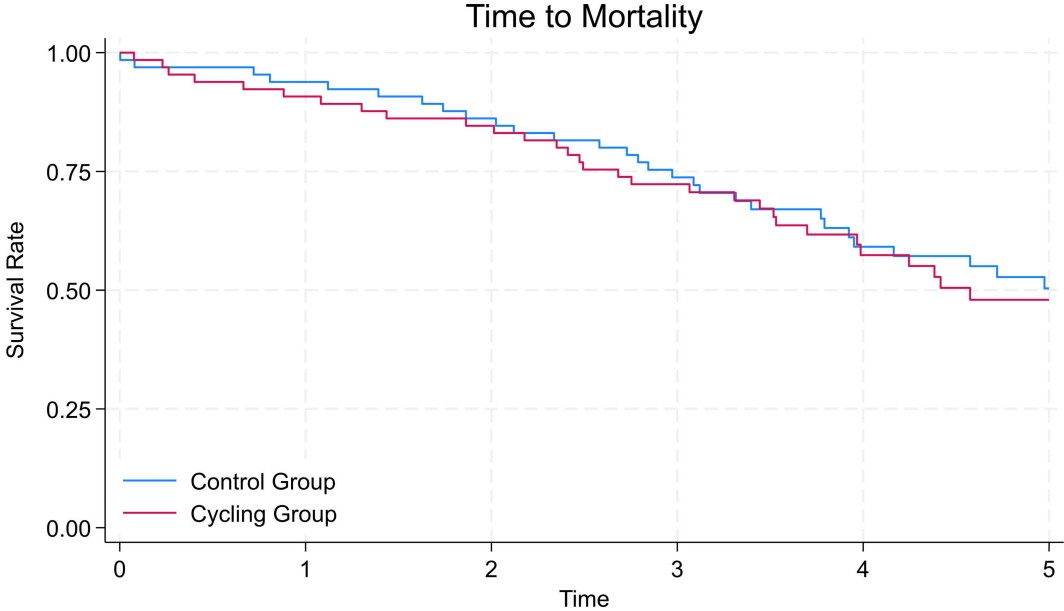

**Fig 1. Kaplan-Meier curve for all-cause mortality in the control and intradialytic cycling groups after five-years follow-up.**

**Table 3. Unadjusted and adjusted cox regression model estimates for all-cause mortality and the composite outcome of cardiovascular events (cardiovascular death or cardiovascular hospitalisations) at five-years follow-up.**

|  | Hazard ratio (95% confidence interval) | *P*-value |
|---|---|---|
| Mortality |  |  |
| All-cause mortality (unadjusted) | 1.09 (0.68–1.76) | 0.71 |
| All-cause mortality, adjusted for: |  |  |
| Cycling group | 1.22 (0.74-2.01) | 0.43 |
| Age (years) | 1.04 (1.01-1.06) | **<0.01** |
| Female | 0.76 (0.40-1.45) | 0.41 |
| Diabetes | 1.66 (1.02-2.68) | **0.04** |
| Dialysis vintage (years) | 0.93 (0.85-1.03) | 0.16 |
| Transplant* | 0.71 (0.31-1.64) | 0.42 |
| Cardiovascular event composite outcome (time to cardiovascular death or cardiovascular hospitalisation) |  |  |
| First cardiovascular event (unadjusted) | 1.39 (0.79-2.46) | 0.26 |
| First cardiovascular event, adjusted for: |  |  |
| Exercise group | 1.52 (0.84-2.72) | 0.16 |
| Age (years) | 1.02 (0.99-1.05) | **0.08** |
| Female | 1.19 (0.60-2.36) | 0.61 |
| Diabetes | 1.89 (1.05-3.42) | **0.03** |
| Dialysis vintage (years) | 0.99 (0.89-1.12) | 0.99 |
| Transplant* | 0.18 (0.04-0.78) | **0.02** |

The reference category for IDC group, diabetes and transplant is 'no'.

*Transplant is a time varying covariate, therefore the interpretation is that receiving a transplant corresponds to a change in the risk of death compared to the time before transplant.

Table 4. Number of hospitalisations and length of stay for hospitalisations at five-years follow-up.

| Hospitalisations | Control (*n*=65) | Intradialytic cycling (IDC) group (*n*=65) | Total (*n*=130) | Coefficient (95% CI) | P-value |
|---|---|---|---|---|---|
| All-cause (Unadjusted)* | 566 | 472 | 1038 | 0.85 (0.66-1.09) | 0.2 |
|  |  |  |  |  |  |
| Adjusted* |  |  |  |  |  |
| IDC group |  |  |  | 0.86 (0.67-1.11) | 0.25 |
| Age (years) |  |  |  | 1.00 (0.99-1.01) | 0.38 |
| Female |  |  |  | 1.02 (0.77-1.35) | 0.9 |
| Diabetes |  |  |  | 1.18 (0.91-1.53) | 0.22 |
| Dialysis vintage (years) |  |  |  | 0.95 (0.90-1.01) | **0.08** |
| Transplant |  |  |  | 1.08 (0.79-1.48) | 0.62 |
| Cardiovascular hospitalisations (Unadjusted)* | 35 | 24 | 59 | 0.70 (0.35-1.37) | 0.3 |
|  |  |  |  |  |  |
| Adjusted* |  |  |  |  |  |
| IDC group |  |  |  |  |  |
| Age (years) |  |  |  | 0.64 (0.33-1.21) | 0.17 |
| Female |  |  |  | 1.00 (0.97-1.03) | 0.9 |
| Diabetes |  |  |  | 1.37 (0.65-2.84) | 0.41 |
| Dialysis vintage (years) |  |  |  | 1.19 (0.62-2.26) | 0.61 |
| Transplant |  |  |  | 0.96 (0.83-1.11) | 0.58 |
|  |  |  |  | 0.14 (0.04-0.45) | **<0.01** |
| Length of stay (hours) |  |  |  |  |  |
| All-cause (Unadjusted)** | 85,510 (56%) | 66,076 (44%) | 1,51,586 | 0.93 (0.67–1.28) | 0.64 |
|  |  |  |  |  |  |
| Adjusted** |  |  |  |  |  |
| IDC group |  |  |  | 1.10 (0.83-1.48) | 0.5 |
| Age (years) |  |  |  | 1.01 (0.99-1.02) | 0.11 |
| Female |  |  |  | 0.87 (0.63-1.20) | 0.4 |
| Diabetes |  |  |  | 1.82 (1.35-2.47) | **<0.01** |
| Dialysis vintage (years) |  |  |  | 0.94 (0.89-0.99) | **0.04** |
| Transplant |  |  |  | 1.08 (0.74-1.58) | 0.67 |
| Cardiovascular (Unadjusted)** | 15,732 (71%) | 6,348 (29%) | 22,080 | 0.59 (0.19-1.74) | 0.34 |
|  |  |  |  |  |  |
| Adjusted** |  |  |  |  |  |
| IDC group |  |  |  | 0.86 (0.21-3.45) | 0.83 |
| Age (years) |  |  |  | 1.04 (0.99-1.10) | 0.1 |
| Female |  |  |  | 0.57 (0.11-2.86) | 0.49 |
| Diabetes |  |  |  | 1.07 (0.25-4.60) | 0.93 |
| Dialysis vintage (years) |  |  |  | 0.91 (0.58-1.43) | 0.69 |

*Negative binomial regression model.

** Generalised linear model gamma distribution, log link.

increased risk associated with diabetes. The population had a five-year mortality of 55%, and 18% of deaths were attributable to a cardiovascular cause, which aligns with expectations for a dialysis population [2,17].

It is known that people receiving haemodialysis are physically inactive [18,19], and these low levels of physical activity strongly associate with cardiovascular disease and mortality [8,20]. This post-trial analysis (which was not powered

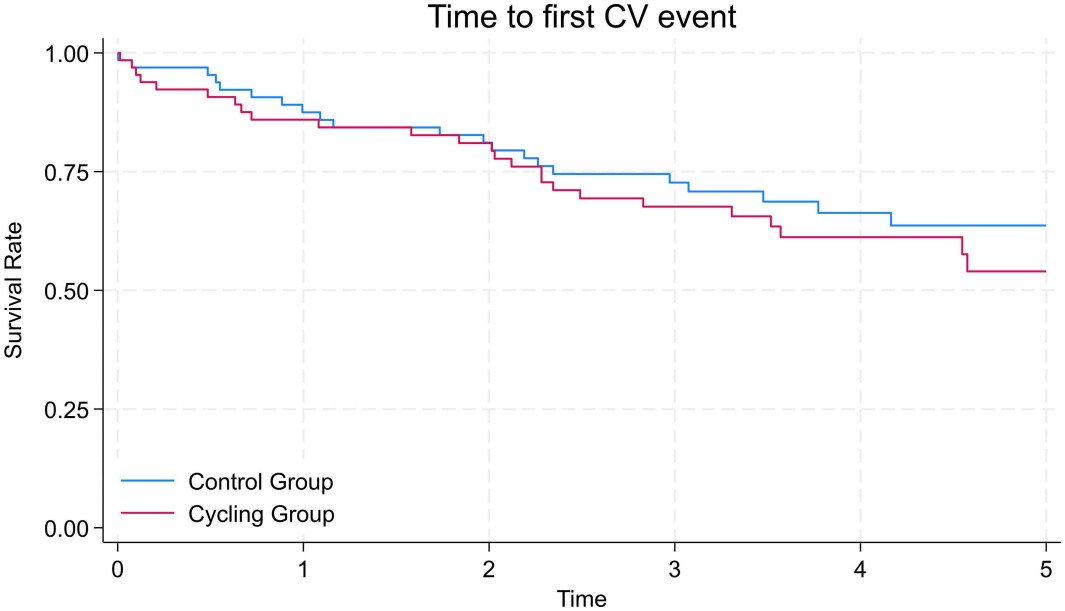

**Fig 2. Kaplan-Meier curve for time to first cardiovascular event (Cardiovascular death or Cardiovascular hospitalisation) in the control and intradialytic cycling groups after five-years follow-up.**

for the presented outcomes) did not observe any effect of a six-month programme of IDC on clinical outcomes. It may be the case that a six-month programme is not a sufficient exercise stimulus to confer long-term benefit on clinical outcomes. However, this appears in contrast to results from the EXCITE trial [11], in their primary trial analysis they showed the benefits of a six-month home walking exercise programme in 227 participants (exercise, $n = 104$; control, $n = 123$), on six-minute walking distance, and the five times sit-to-stand test [21]. In addition, they have since reported that being randomised to the intervention was associated with benefits which persisted in the post-trial period [11,12]. Moreover, they reported that taking part in the programme was associated with a 29% reduction in their composite primary outcome of hospitalisation and mortality (HR 0.71, 95% CI: 0.50, 1.00, $p = 0.06$) at 36 months post-trial [11]. When this study assessed mortality alone, there was no significant difference between groups, highlighting that their observed changes were driven by a significant reduction in hospitalisations [11]. Similarly, the recently reported DIATT trial [10], the largest trial in this area, which randomised 1211 individuals to a 12-month programme of intradialytic exercise or a usual care control group reported a significant reduction in hospitalisations ($p = 0.024$), and days in hospital ($p = 0.036$), between groups at the end of the programme. In addition, another recent trial [22], reported a significant reduction (HR 0.17, 95% CI: 0.04, 0.8; $p = 0.02$) in mortality at 12 months following the completion of a six-month programme of intradialytic exercise. Although it is worthy of note here that the reported confidence intervals for the HR are large (0.04–0.8) which indicates uncertainty around this estimate, furthermore the sample size calculation of 74 participants (exercise, n = 37; control, n = 37) was based on an assumed reduction of 75% (a HR of 0.25) in mortality following the intervention. A 75% reduction in mortality is a large effect size especially for a non- pharmacological intervention, particularly given the 29% reduction in the composite outcome of hospitalisation and mortality reported in the EXCITE trial [11], which is more realistic. Taken together, we believe that the results of the latter trial [22] should be interpreted with caution.

Within our data we report a non-significant 9% reduction in number of hospitalisations, which is in agreement with our previously reported data for the same trial (and findings from the EXCITE and DIATT trials), which showed the IDC intervention to be cost-effective; primarily driven by a reduction in hospitalisations [23]. It is likely that this study was underpowered to detect significant differences in number of hospitalisations, as the CYCLE-HD trial was powered for changes

**Table 4.** (Continued)

in left ventricular mass. Using our data, a sample size calculation based on a 9% reduction in hospitalisations suggests that we would need to recruit 955 participants in total to detect a significant difference in hospitalisations due to the effect of a six-month IDC intervention. Notably, this is a comparable number of participants recruited to the DIATT trial (which randomised 1,211 people), and reported a significant reduction in hospitalisation per patient following a 12-month intradialytic exercise programme. The numerical reduction in the composite outcome in the EXCITE trial [11] and the significant reductions in hospitalisations reported by the both the EXCITE and DIATT trials [10,11] (compared to the data we present here for the CYCLE-HD trial) may be explained by a longer intervention period [10], larger samples sizes [10,11], or being provided with an exercise diary, and encouraged to continue the intervention in the post-trial period [11]. Indeed, in the EXCITE trial, when analysis was focussed on adherence to the exercise intervention during (and in) the post-trial period, those with high adherence (55/104 (53%) participants in the exercise group) had a statistically significant reduction in the composite primary outcome and the secondary outcome of hospitalisation alone compared to the low adherence or control group. In contrast, following the six-month IDC intervention in CYCLE-HD, the programme was not offered thereafter to participants (they did not continue to exercise), and we do not know whether participants were physically active in their time outside of dialysis. This possible difference in behaviour may partly explain the non-significant findings for mortality and hospitalisation at five years in the CYCLE-HD trial compared to the EXCITE trial.

Despite the evidence from these previous trials on the effect of programmes of exercise in the haemodialysis population on hospitalisation and mortality outcomes [10,11], there still remains no appropriately powered trial investigating the effect on clinical outcomes such as mortality and cardiovascular events, which has been highlighted by the latest UK Kidney Association Clinical Practice Guideline for Exercise and Lifestyle in CKD [7]. The lack of this data (alongside other barriers) has hindered the implementation of exercise programmes for people with CKD. Along with trial data, consideration needs to be given to implementing and integrating sustainable programmes of exercise into routine patient care, which would likely involve dedicated resource allocation, multidisciplinary team working, and addressing patient and healthcare professional activation, and behaviour change.

The major strength of this study is that it is one of the few in this population that has investigated long-term follow-up of an exercise trial on clinical outcomes. The primary limitations of this study are the small sample size and post-hoc design, as the original trial was powered to show changes in left ventricular mass and not long-term clinical outcomes such as mortality and hospitalisation. The participants did not continue the IDC intervention following the six-month initial trial, so we are unable to comment on the long-term effects of sustained IDC. Furthermore, as the data for this study is observational follow-up of an RCT, we have no data for other important factors that may affect mortality and hospitalisation in these individuals. Lastly, hospitalisations were captured from the University Hospitals of Leicester through the HES database, therefore there may be missing data as a result of hospitalisations occurring outside of this National Health Service Trust.

## Conclusion

Our study demonstrated no significant benefit in mortality and hospitalisations at five years following a six-month programme of IDC exercise. Whilst this study was not powered for these outcomes, the reduction in hospitalisations was similar to that seen in the DIATT trial (although the collection time is different), and the lack of significance may be a result of being underpowered. Further evidence is required to investigate the effect of more sustained, and incorporated exercise on clinical outcomes in this population.

## Supporting information

**S1 Table. ICD-10 codes used to define hospitalisations related to cardiovascular diagnoses.**
(DOCX)

## Author contributions

**Conceptualization:** James O. Burton, Daniel S. March.

**Data curation:** Daniel Lawday, Darren R. Churchward, Patrick J. Highton, Hannah M. L. Young, Matthew P. M. Graham-Brown, James O. Burton, Daniel S. March.

**Formal analysis:** Hannah M. Worboys, Laura J. Gray, James O. Burton, Daniel S. March.

**Funding acquisition:** James O. Burton.

**Investigation:** Sherna F. Adenwalla, Darren R. Churchward, Hannah M. L. Young, Matthew P. M. Graham-Brown, James O. Burton, Daniel S. March.

**Methodology:** Sherna F. Adenwalla, Laura J. Gray, Katherine L. Hull, Darren R. Churchward, Matthew P. M. Graham-Brown, James O. Burton, Daniel S. March.

**Project administration:** Katherine L. Hull, Matthew P. M. Graham-Brown, James O. Burton, Daniel S. March.

**Supervision:** James O. Burton, Daniel S. March.

**Writing – original draft:** Sherna F. Adenwalla, Hannah M. Worboys, Laura J. Gray, Katherine L. Hull, Darren R. Churchward, Patrick J. Highton, Hannah M. L. Young, James O. Burton, Daniel S. March.

**Writing – review & editing:** Sherna F. Adenwalla, Hannah M. Worboys, Daniel Lawday, Laura J. Gray, Katherine L. Hull, Darren R. Churchward, Patrick J. Highton, Hannah M. L. Young, Matthew P. M. Graham-Brown, James O. Burton, Daniel S. March.

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
