## [Decision Letter · Decision Letter 0]

29 Jul 2025

PONE-D-25-28929The effect of a six-month programme of intradialytic cycling on survival and hospitalisations in patients requiring haemodialysis: 5-year follow-up of the CYCLE-HD randomised controlled trialPLOS ONE

Dear Dr.  March,

Thank you for submitting your manuscript to PLOS ONE. After careful consideration, we feel that it has merit but does not fully meet PLOS ONE’s publication criteria as it currently stands. Therefore, we invite you to submit a revised version of the manuscript that addresses the points raised during the review process.

We look forward to receiving your revised manuscript.

Kind regards,

Fan Zhang

Academic Editor

PLOS ONE

Journal Requirements:

3. In the online submission form, you indicated that “Deidentified individual participant data collected for the study, and a data dictionary defining each field in the set, will be made available to others on specific to request to the study chief investigator (JOB), and corresponding author (DSM) provided all regulatory approvals are met.”

6. We note you have included a table to which you do not refer in the text of your manuscript. Please ensure that you refer to Table 1 in your text; if accepted, production will need this reference to link the reader to the Table.

7.If the reviewer comments include a recommendation to cite specific previously published works, please review and evaluate these publications to determine whether they are relevant and should be cited. There is no requirement to cite these works unless the editor has indicated otherwise. 

Reviewers' comments:

Reviewer's Responses to Questions

**Comments to the Author**

1. Is the manuscript technically sound, and do the data support the conclusions?

Reviewer #1: Yes

Reviewer #2: Yes

Reviewer #3: Yes

2. Has the statistical analysis been performed appropriately and rigorously? 

Reviewer #1: Yes

Reviewer #2: Yes

Reviewer #3: Yes

3. Have the authors made all data underlying the findings in their manuscript fully available?

Reviewer #1: Yes

Reviewer #2: Yes

Reviewer #3: Yes

4. Is the manuscript presented in an intelligible fashion and written in standard English?

Reviewer #1: Yes

Reviewer #2: Yes

Reviewer #3: Yes

5. Review Comments to the Author

Reviewer #1: It is a nice article titled - The effect of a six-month programme of intradialytic cycling on survival and

hospitalisations in patients requiring hemodialysis " I find there are some grammatical errors which will need corrected but else find it acceptable

Reviewer #2: This manuscript addresses an important and timely public health issue by examining the association between living alone and self-rated health among older adults in a local Japanese municipality. The focus on gender differences adds value and relevance to the research question.

The study design and statistical methods (particularly the use of logistic regression analysis stratified by gender) are appropriate and clearly reported. The discussion appropriately contextualizes the findings with previous literature, and the authors have taken care to acknowledge the limitations of the cross-sectional design.

I appreciate the clarity of the writing and the careful attention to detail throughout the manuscript. The conclusions are supported by the results, and the implications for social support and policy are well-articulated.

I have no major concerns. I recommend the manuscript for publication with no further revisions.

Reviewer #3: This is a long-term follow-study of the CYCLE-HD trial, which was a 6-month randomized controlled trial of intra-dialytic cycling. The authors examine the effect of the intervention on all-cause mortality, hospitalizations, and cardiovascular events at 5 years. There were no statistically significant effects on mortality, a composite outcome of cardiovascular mortality and hospitalizations, or other outcomes. The authors do a good job of placing their findings in the context of prior studies in a thoughtfully written discussion. They also acknowledge that CYCLE-HD, which randomized 130 participants, was underpowered for the purposes of this analysis. I have the following comments:

1. It is not clear why time zero was defined differently for the intervention and control arms (lines 96-97). The time of randomization would seem to be the appropriate starting time for all participants.

2. The text implies that hospitalizations were only captured for University Hospitals of Leicester (line 99). If so, there is the possibility of missing data due to external hospitalizations. This should be added as a limitation.

3. Results, lines 142-146: this text presumably refers to events occurring between the start of the study and the end of follow-up, not within the 6 months of the intervention. This should be clarified.

4. Line 277: Given that the effect size of reduction in hospitalizations was similar to that seen in DIATT, the lack of statistical significance may simply be due to lack of power, and does not indicate absence of clinical significance. Thus “unlikely to be beneficial” may be too strong of a statement.

5. Table 1: both rows for mortality have incorrect values for percentages (controls, all-cause mortality 57%; IDC, CV deaths 22%).

6. There are multiple typos or missing callouts throughout the manuscript (e.g. lines 139-140).

7. Line 198: “physically inactive”

6. PLOS authors have the option to publish the peer review history of their article (what does this mean? ). If published, this will include your full peer review and any attached files.

**Do you want your identity to be public for this peer review?** For information about this choice, including consent withdrawal, please see our Privacy Policy .

Reviewer #1: No

Reviewer #2: No

Reviewer #3: No

---

## [Author Response · Author response to Decision Letter 1]

6 Aug 2025

We have attached all responses to Editor and Reviewer comments as a file. We do not believe that comments from Reviewer 2 relate to our manuscript. Could the Editor provide clarification around this please.

---

## [Editor Report · Decision Letter 1]

1 Sep 2025

The effect of a six-month programme of intradialytic cycling on survival and hospitalisations in people requiring haemodialysis: 5-year follow-up of the CYCLE-HD randomised controlled trial

PONE-D-25-28929R1

Dear Dr. 

We’re pleased to inform you that your manuscript has been judged scientifically suitable for publication and will be formally accepted for publication once it meets all outstanding technical requirements.

Kind regards,

Fan Zhang

Academic Editor

PLOS ONE
---

## [Editor Report · Acceptance letter]

PONE-D-25-28929R1

PLOS ONE

Dear Dr. March,

I'm pleased to inform you that your manuscript has been deemed suitable for publication in PLOS ONE. Congratulations! Your manuscript is now being handed over to our production team.

Kind regards,

on behalf of

Dr. Fan Zhang

Academic Editor

PLOS ONE